# Agrivoltaics: Modeling the relative importance of longwave radiation from solar panels

**Laurel A. Shepard** *⬮, **Chad W. Higgins** *⬮, **Kyle W. Proctor**

Department of Biological and Ecological Engineering, Oregon State University, Corvallis, Oregon, United States of America

⬮ These authors contributed equally to this work.
* sheparla@oregonstate.edu (LAS); chad.higgins@oregonstate.edu (CWH)

**Data Availability Statement:** All relevant data are within the manuscript and its Supporting information files.

## Abstract

Agrivoltaics, which integrate photovoltaic power production with agriculture in the same plot of land, have the potential to reduce land competition, reduce crop irrigation, and increase solar panel efficiency. To optimize agrivoltaic systems for crop growth, energy pathways must be characterized. While the solar panels shade the crops, they also emit longwave radiation and partially block the ground from downwelling longwave radiation. A deeper understanding of the spatial variation in incoming energy would enable controlled allocation of energy in the design of agrivoltaic systems. The model also demonstrates that longwave energy should not be neglected when considering a full energy balance on the soil under solar panels.

## Introduction

The climate crisis has put an urgency on the transition from fossil fuels to clean, renewable energy. Meanwhile, population growth and soil degradation create need for new cropland. Growing demand for food and clean energy has led to competition between croplands and solar arrays for land. An estimated 6000 TWh of PV power will be generated in 2050 [1], most of which can be met with building integrated PV and rooftop PV. The remaining demand can be met with land-based solar farms [1]. To mitigate competition, some land-based solar farms could be converted to agrivoltaic systems, in which crops are grown under solar panels. Agrivoltaics simply refers to land where both solar panels and agriculture are present. This can take many forms, including rows of crops grown in the space between panels, panels on top of a greenhouse, or even livestock grazing around the panels.

Agrivoltaics offers many benefits. One model predicted a 35–75% increase in global land productivity based on Land Equivalent Ratios that sum the effects of both crop yield and electricity [2]. Furthermore, co-developing land for both solar power and agriculture could supply 20% of the total electricity generation in the U.S. using less than 1% of the annual US budget [3]. In addition to being economically viable, agrivoltaics may be especially effective for shade tolerant crops. Some crops increase their yield when grown in the shade, especially in hot and dry conditions. For example, Weselek et al. found that during a hot and dry summer, agrivoltaics increased crop yields of winter wheat and potato by 2.7% and 11%, respectively [4].

**Funding:** The authors received no specific funding for this work.

**Competing interests:** The authors have declared that no competing interests exist.

Another experimental farm with solar panels in a stilt-mounted system saw a 5.6% yield increase in corn, a typically shade-intolerant crop [5]. Agrivoltaics also has the potential to reduce crop water demand due to increased water productivity, even for shade-intolerant crops.

However, agrivoltaics has its limitations. There is a trade-off between crop yield and electricity generation. Unsurprisingly, many plants produce less biomass in the shade, assuming they are not limited by water. This may lead to lower crop yields under solar panels, which Weselek observed for winter wheat, potato, and grass-clover [4]. An agrivoltaic system in Japan yielded only 80% of the rice yield in the control [6]. Several models have demonstrated that a shorter distance between panel rows reduces crop yields [7, 8]. While shade-tolerant crops may not have reduced yields, there is limited data to help identify those crops. Crop quality may also be affected; Marrou (2013) found reduced leaf emission rates for lettuce and cucumber in the juvenile phase when grown under solar panels [9]. Solar panels can also obstruct the path for farm equipment, and overhanging foliage can partially block the irradiation to the panels.

While there is high potential for agrivoltaics, maximizing its performance will require a deeper understanding of the underlying physical processes. Models can be used to help predict the reduction in crop yield, to maximize crop yield, to minimize heterogeneity in soil moisture, or to optimize the system in other ways. There is still much to learn, including the best crop species for agrivoltaics, how to optimize solar panel configurations, and to what extent crop evapotranspiration cools off the panels. The heterogeneity of longwave radiation at the ground surface has not been widely explored.

All mass emits longwave radiation according to its temperature, including solar panels and the air itself. This radiation continually adds energy to the ground surface. This paper develops a model to quantify the downwelling longwave energy at the ground surface in an agrivoltaic array. First, geometry and the first law of thermodynamics are applied to model the heat flux resulting from a solar panel's longwave radiation. Next, we adapt the model to include the solar panel dimensions and weather data specific to a published case study (Adeh et al. 2018) [10]. We then analyze the longwave, shortwave, and total radiative heat flux across time and space at the land surface at both the control and agrivoltaic areas. Model results indicate that the longwave and shortwave radiative fluxes are of similar order of magnitudes under some conditions, even during the daytime.

## Methods

The heat flux through a point location on the ground is a result of shortwave and downwelling longwave radiation. Because rows of solar panels are identical along their length, and the row width is much greater than the distance between rows, edge effects can be ignored. Therefore, this is a 2D problem concerned with only a cross section of a solar array. The model also contains the following assumptions:

1. Every row of panels is infinitely long

2. Missing weather parameters for small time steps (less than 1 day) were approximated via interpolation.

3. The geometry is based on panels that slope upward in the positive x-direction.

4. The estimation of downwelling longwave radiation assumes clear skies.

5. Assumptions made in the energy balance on a panel derived from measured weather parameters:

a. The panel temperature was assumed to be exactly equal to $T_P$ (i.e. the energy balance accounts for all factors contributing to panel temperature).

b. $T_g = T_{air}$ for all times and positions.

c. The panels are perfect black bodies (i.e. have an emissivity of 1.0)

d. Longwave radiation from surrounding panels is negligible.

e. Electrical efficiency of the panels follows Eq 15 even when $|T_P - T_{ref}| > 20$

Fig 1 summarizes the methods. At a given time $t$, the weather data enables computation of the panel temperature and longwave radiation from the sky. Using the panel temperature and

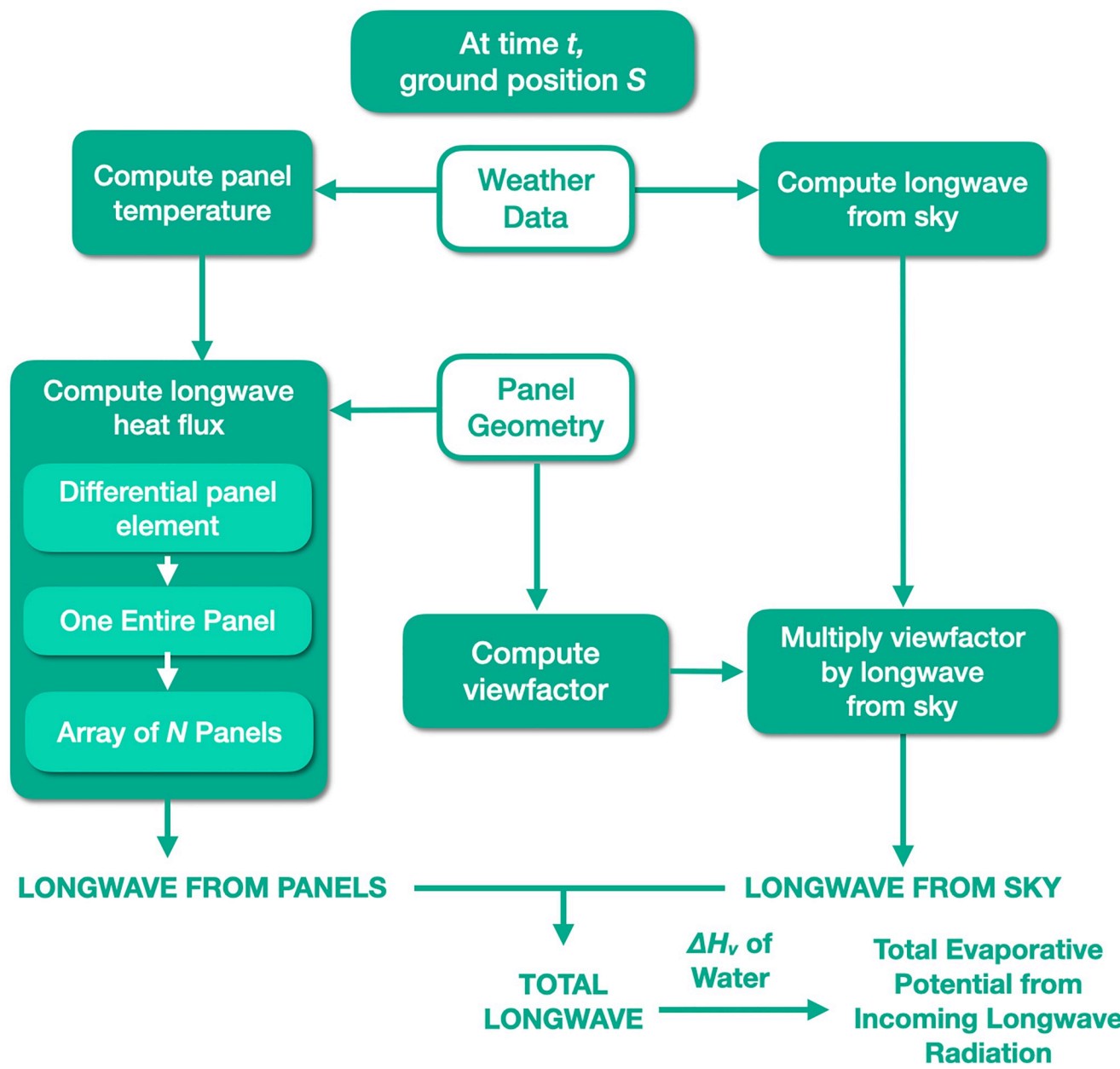

**Fig 1. Complete process for modeling evaporation potential from longwave energy.**

geometry, the longwave radiative heat flux arriving at ground position $S$ is composed of the longwave radiation from the panels and the longwave radiation from the sky. Flux from the panels is integrated from a differential panel element, to one entire panel, and finally to the entire solar array. The heat flux from the sky that $S$ receives is the longwave radiation from the sky times the viewfactor at $S$ (viewfactors are dependent on panel geometry only). Finally, we sum the two sources of heat flux and use latent heat of vaporization to generate the evaporation potential at time $t$ and position $S$.

## Differential panel element

At its core, the total heat flux at a given ground position is the sum of the fluxes from many individual panel elements. The heat flux through a given ground position $X$, as a result of a *single* differential element of a panel, can be described by $q_e$. It can be imagined that the total energy leaving the element in all directions must be equal to the energy leaving a circular boundary that intersects the point of interest on the ground, as demonstrated in Fig 2a. To obtain an expression for $q_e$, it begins with the energy balance in Eq (i). This states that the product of the flux from the panel and the area of a panel element is equal to the product of $q_e$ and $A_{p \to g}$, where $A_{p \to g}$ is the area of the circular boundary.

$$q_p A_e = q_e A_{p \to g} \tag{i}$$

The area of a panel element $A_e$ is equivalent to the differential length $d\ell$ times the width of the array (i.e. width into the page $W$). The variable $A_{p \to g}$ represents half the area of the side of a cylinder, where the cylinder represents the outgoing radiation from a single panel element. The cylinder's intersection with the ground represents where the radiation strikes the ground. The outgoing radiation is constricted to a half-cylinder due to the adjacent panel elements (Fig 2a). This cylinder has length $W$ and a radius equal to the distance from the panel element $x'$ to ground position $X$. This radius can be found by the Pythagorean theorem, where the legs of the triangle are the height of the panel element $H$ and the horizontal distance $(X - x')$. The flux $q_p$ is simply found by the blackbody radiation equation, $q = \sigma T^4$. Eq (ii) is the result of substituting these expressions for $q_p$, $A_e$, and $A_{p \to g}$ into Eq (i), which can then be solved for $q_e$.

$$\sigma T_p^4 (d\ell \cdot W) = q_e \left( W \cdot \frac{1}{2} \cdot 2\pi \sqrt{H^2 + (X - x')^2} \right) \tag{ii}$$

Rearranging Eq (ii) for $q_e$ results in Eq 1.

$$q_e(X) = \frac{\sigma T_p^4 \, d\ell}{\pi \sqrt{H^2 + (X - x')^2}} \tag{1}$$

where $\sigma$ is the Stefan-Boltzmann constant ($5.67 \cdot 10^{-8} \, kg \, s^{-1} \, K^{-4}$), $T_P$ is the temperature of the panel, $h$ is the height of the panel element, $X$ is the horizontal length from the lower edge of the panel to the ground position of interest, and $x'$ is the horizontal length from the lower edge of the panel to the differential element (Fig 2a). The height $H$ of the differential element can be defined as:

$$H = H(x') = H_0 + x' \tan\theta \tag{2}$$

where $h_0$ is the height of the lower edge of the panel, and $\theta$ is the angle of the panel.

The flux at ground position X from an individual panel element ($q_e$) was computed numerically. Eqs 1 and 2 were evaluated for the constants $x'$, $h_0$, $\theta$, $T_P$, and the independent variable $X$.

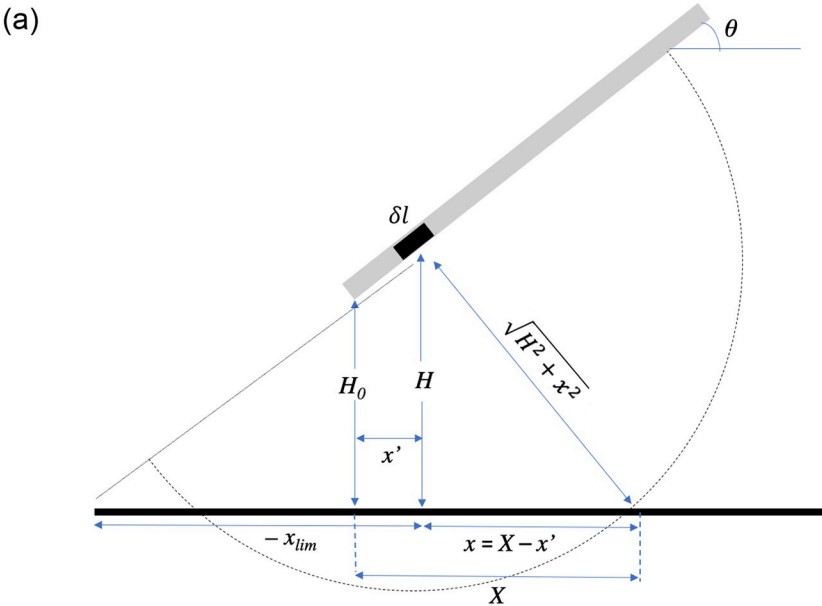

(a)

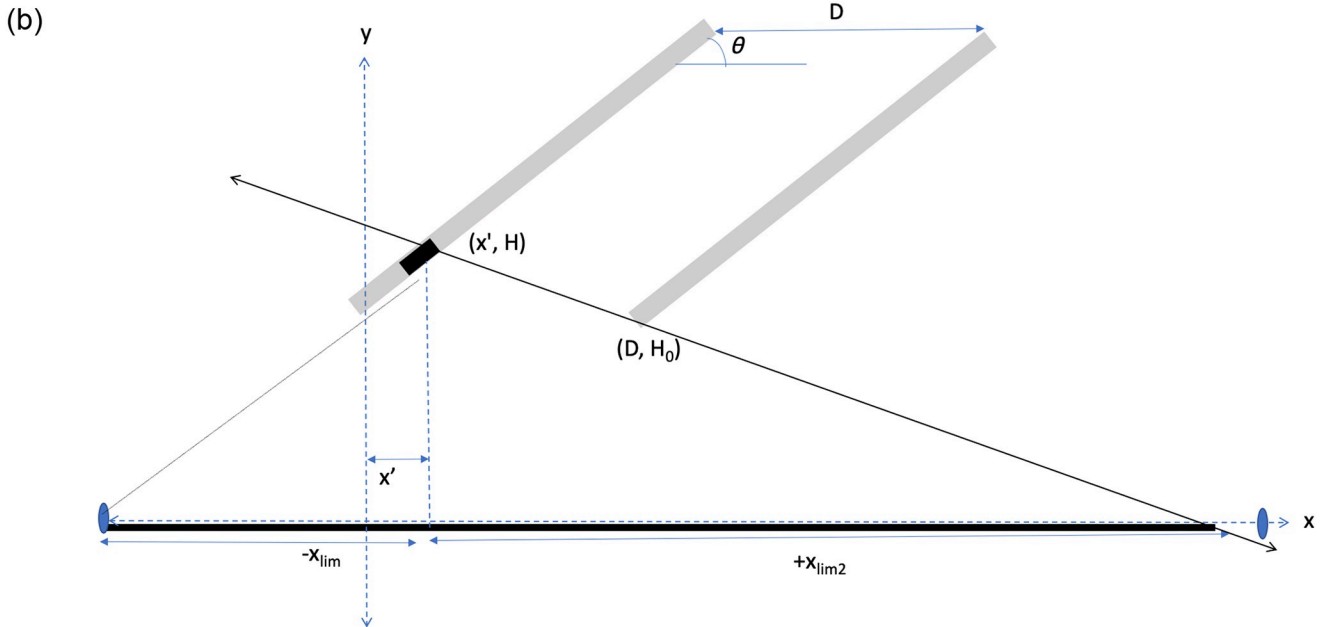

(b)

**Fig 2.** a. Heat flux from a differential panel element. b. Limits of influence for a given panel element (Not to scale with case study array).

Note that $x'$ is a constant because it represents a single panel element, but $X$ is an array of ground positions across the entire solar array that will each have unique flux values. The result is an array of values for $q_e$ at each $X$. The range of $X$ is bound. For $X < 0$, the range extends to $x_{lim}$, which is the point on the ground if one were to extend the panel to touch the ground (See $x_{lim}$ on Fig 2a, 2b). For $X < -x_{lim}$, the radiation is blocked by the lower panel elements.

For X > 0, the ground positions can become shielded from panel element $x'$ by the adjacent panel (Fig 2b). The place where this happens, $x_{lim2}$, is a function of the distance between two panels, $D$. $x_{lim2}$ can be found by defining a line with two points: the panel element and the bottom of the adjacent panel. $x_{lim2}$ is the x-intercept of that line.

$$x_{lim2} = -H \cdot \frac{H - H_0}{x' - D} + x' \tag{3}$$

Since $x_{lim2}$ is dependent on the panel element position $x'$, it was computed concurrently with the flux calculations above. The flux $q_{element}$ was then computed for all X values between $x_{lim}$ and $x_{lim2}$. In the special case where the panel element was the lowest element on the panel (h = $h_0$), there would be nothing to block the radiation from extending infinitely in all directions, and were accounted for in the flux calculations.

Fig 3a depicts the heat fluxes resulting from three panel element locations. 'Low end' corresponds to the special case where the panel element is at the lower end of the panel; 'middle end' corresponds to the panel element in the approximate middle of the panel; and 'high end' corresponds to the panel element at the upper end of the panel.

## Flux from one entire panel

The flux from one entire panel was determined by integrating Eq 1 across the length of the panel (i.e. from $x' = 0$ to $x' = L \cos\theta$). The integration was done numerically by discretizing the full panel length into 100 elements each of equal length $d\ell = 0.01$ m. The result is a heat flux distribution from an entire panel for a given ground position X. Recall that $x'$ is the horizontal position of a given panel element, so its maximum value is $L \cos \theta$ for a panel of length $L$. The heat flux to the ground as a result of the entire panel is defined in Eq 4.

$$q_{P,X} = \int_0^{L \cos\theta} \frac{\sigma T_P^{\;4}}{\pi \sqrt{[H(x')]^2 + (X - x')^2}} \, dx' \tag{4}$$

where $x' = 0$ and $X = 0$ occur at the lower edge of the panel of interest. Fig 3b shows the resulting flux from one single panel, given the inputs defined in Table 1.

## Flux from entire array

An array-scale coordinate system, S, must be defined because each panel is expressed in a relative coordinate system, X. There is overlap between each panel's influence on the ground (i.e. ground positions receive longwave radiation from multiple panels). Therefore, the effect of the entire array on a given position is defined in Eq 5.

$$q\{S\} = \sum_{i=1}^{N} q\{S - D(i - 1)\} \tag{5}$$

where $N$ = the number of panels in the array, $i$ = the $i^{th}$ panel in the array, $D$ = horizontal distance between two adjacent panels, and $S$, the position of interest, is a constant within the sum. Note that $S$ is the global position within the entire array; $S = 0$ occurs at the lower edge of the leftmost panel, while $X = 0$ occurs at the lower edge of a particular panel of interest, when examining that panel's effect. Eq 5 is evaluated for a discrete set of $S$ positions. The result is plotted for the first 5 panels in Fig 4.

(a)

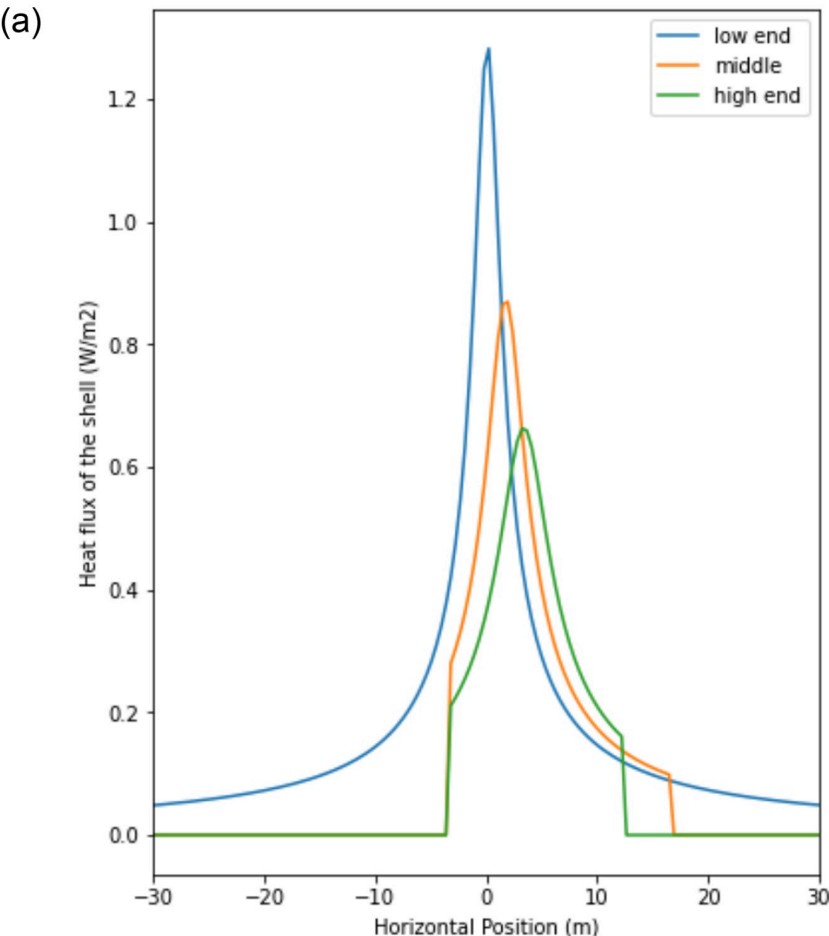

(b)

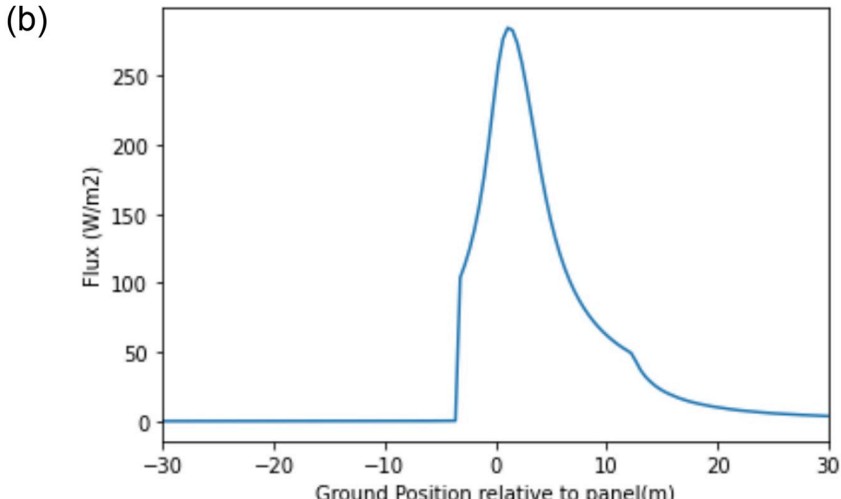

**Fig 3.** a. Heat flux resulting from three different panel elements. b. Heat flux from one single panel, for arbitrary panel temp $T_P$ = 300 K, where $x$ = 0 occurs at lower edge of panel.

Table 1. Panel geometry parameters.

| Symbol and Value | Definition |
|---|---|
| $L$ = 3.56 m | Length of each panel |
| $D$ = 6.2 m | Distance between adjacent panels |
| $\theta$ = 0.314 rad | Panel tilt |
| $H_o$ = 1.1 m | Height of lower edge of panel above the ground |
| $\Delta S$ = 0.1 m | Distance between two ground positions in computational model, given the above inputs |

## Flux from the sky

An additional component in the model account for the longwave radiation from the sky that passes through the gaps between panels. In the absence of panels, the clear sky radiation contribution can be calculated as follows by Brutsaert (1975):

$$L_{sky} = 1.24\sigma\left(\frac{e}{T_{air}}\right)^{\frac{1}{7}} T_{air}^{\;4} \tag{6}$$

where $L_{sky}$ is the downwelling longwave radiation (W/m$^2$), $\sigma$ = 5.67 · 10$^{-8}$ W m$^{-2}$ K$^{-4}$ is the Stefan-Boltzmann constant, $e$ is the vapor pressure of water in air (hPa), and $T_{air}$ is the measured air temperature (K). The vapor pressure, $e$, can be estimated with the saturation vapor

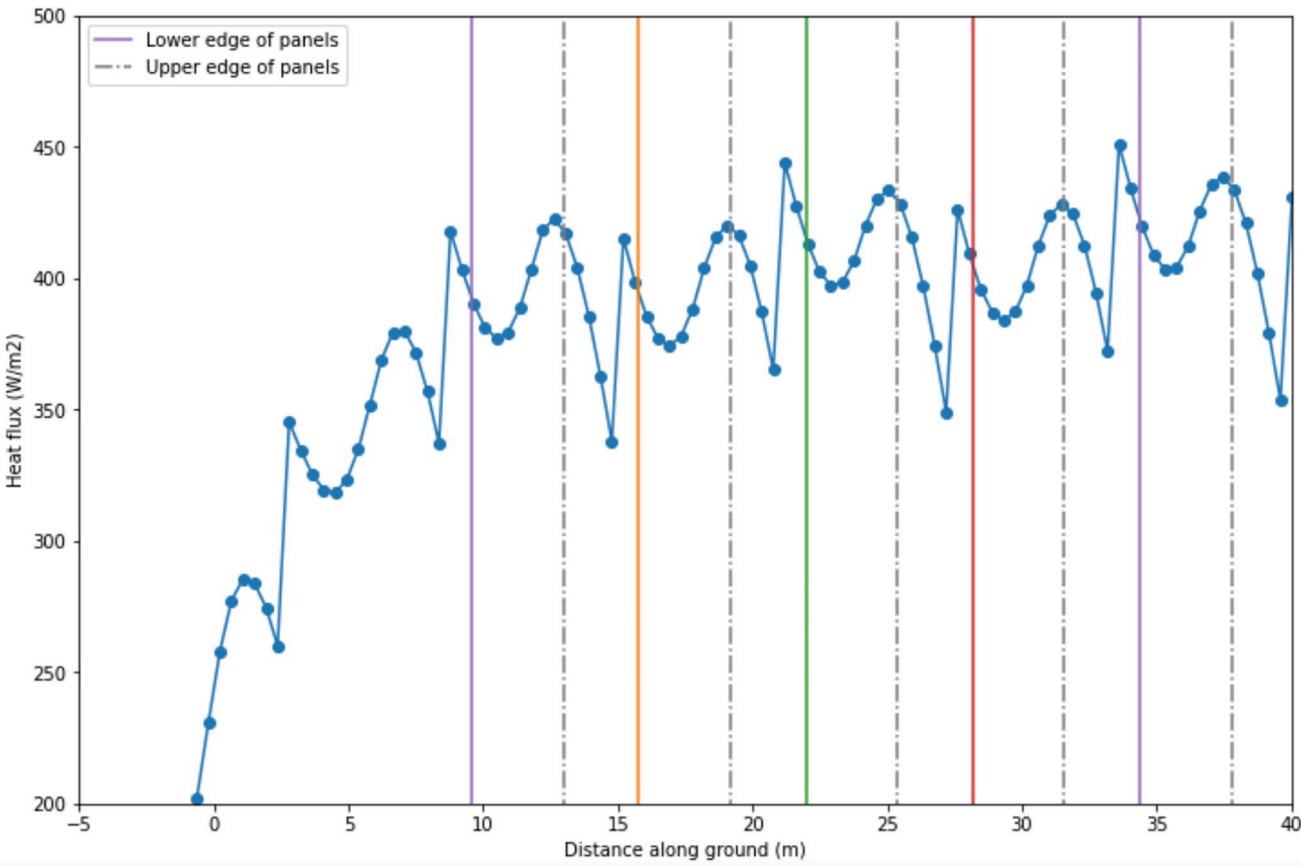

**Fig 4. Heat flux as a result of solar panels only (no sky), for arbitrary panel temp T$_P$ = 300 K, where x = 0 occurs at lower edge of leftmost panel.**

pressure, $e^*$:

$$e = RH \cdot e^* \tag{7}$$

where $RH$ is the relative humidity. The saturation vapor pressure e* can be estimated via the Tetens equation [11]:

$$e^* = 0.611 \exp\left(\frac{17.3 T_{air}}{T_{air} + 237.3}\right) \tag{8}$$

where $T_{air}$ is expressed in ºC and e* is expressed in Pa.

## Flux from the Sky: Viewfactors

The ground beneath the panels is not exposed to clear sky conditions; rather, the extent of exposure is quantified by the view factor, $F_v$ (unitless), at each position. The view factor, in this case, is the fraction of sky that is visible from any given ground position. Thus, the contribution of radiation from the sky at any position is the viewfactor multiplied by the total longwave radiation from the sky found in Eq 6.

The method for computing the view factor at each point is outlined in Fig 5. The viewfactor may be expressed as the superposition of smaller subcomponent views, as defined in Table 2, and illustrated in Fig 6a. The two main sub-views are the views between the ground and panels, and the views between two adjacent panels (wedges a and b, respectively). Concomitantly, the sky blocked by panels may be composed of one or multiple panels. The sum of the fractions of exposed views and the fractions of blocked views must equal 1. Therefore, we may choose to calculate the blocked or the open views. This choice is a matter of convenience. The section below considers the blocked views (wedges c and d).

From Fig 6a, it is clear that the wedges blocked by multiple panels are delineated by a first line connecting the ground position and the upper edge of a panel, so called the "left edge top" and "right edge top" angles ($\alpha_{t_L}$ and $\alpha_{t_R}$). The second line is formed by connecting the ground position and the lower edge of the most distant panel, so-called the left edge angle, $\mu$, and right edge angle, $\gamma$. The second line is easily determined because the particular relevant panel is predetermined; it is always the panel at the outer edge of the array. By contrast, the panel that intersects the first line must be determined for each ground position. This panel will be referred to as the transition panel.

Consider the special case when a ray from a ground position intersects the exact lower edge of the transition panel *and* the upper edge of the next most-distant panel. This line forms an angle with the ground that is termed the critical angle(s). Note that there is a critical angle to the left and to the right of the ground position. This is illustrated in Fig 7a. For an arbitrary ground position, a ray extending from the ground position at the critical angle(s) intersects the transition panel. We use those rays to find the left and right transition panels.

The left and right critical angles are determined from the parallelogram created by two adjacent panels (Fig 7a). The diagonals of the parallelogram, M and G, are calculated with the law of cosines. The left critical angle, $\omega$, and the right critical angle, $\phi$, are found with the law of sines.

These angles, $\omega$ and $\phi$, are used to identify the transition panels for a given ground position. The transition panel is the panel for which the critical angle is in between the top angle ($\alpha_t$) and the bottom angle ($\alpha_b$). The bottom and top angles are found by Eqs 9 and 10, respectively.

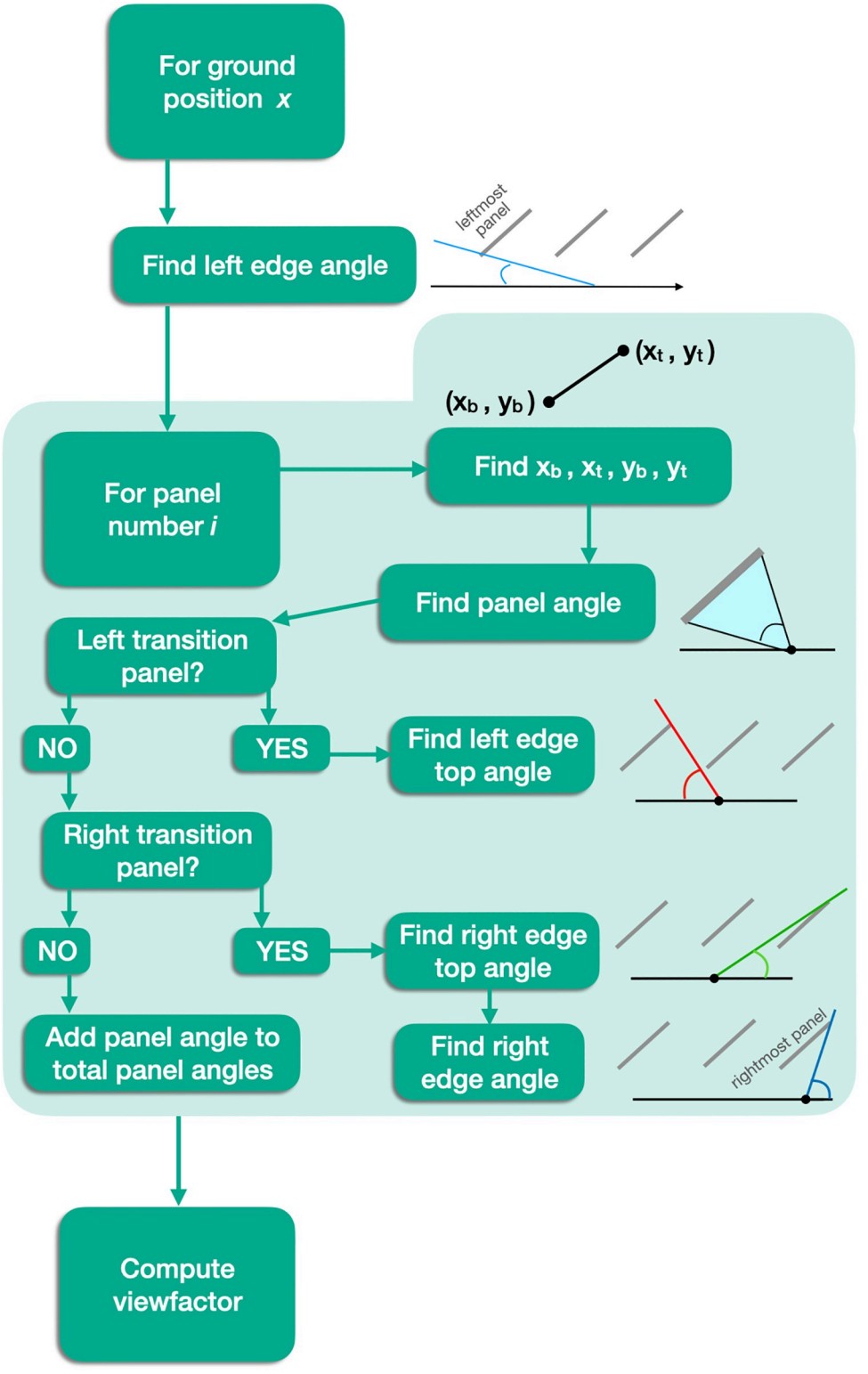

**Fig 5. Computing the view factor.**

**Table 2. Definition of angles within a viewfactor.**

| Variable Name | Symbol | Definition |
|---|---|---|
| Panel tilt | $\theta$ | Tilt of the solar panel itself. |
| Left critical angle | $\omega$ | The angle that identifies the left transition panel. Constant for all ground positions, assuming constant panel geometry. |
| Right critical angle | $\phi$ | The angle that identifies the right transition panel. Constant for all ground positions, assuming constant panel geometry. |
| Top angle | $\alpha_t$ | When a ray is extended from ground position $X$ to the upper edge of panel $i$, the top angle is the angle between this ray and the ground. |
| Bottom angle | $\alpha_b$ | When a ray is extended from ground position $X$ to the lower edge of panel $i$, the bottom angle is the angle between this ray and the ground. |
| Panel angle(s) | $\alpha_t - \alpha_b$ | the angle created by the bottom edge of the panel, the ground position, and the top edge of the panel. Only factored into the viewfactor for the panel(s) between the left and right transition panels. (wedge c) |
| Left edge top angle | $\alpha_{t_L}$ | Angle between the ground and the ray extending from ground position $X$ to the upper edge of the left transition panel |
| Right edge top angle | $\alpha_{t_R}$ | Angle between the ground and the ray extending from ground position $X$ to the upper edge of the right transition panel |
| Left edge angle | $\mu$ | the angle of exposure from the leftmost panel in the solar array (wedge a) |
| Right edge angle | $\gamma$ | the angle of exposure from the rightmost panel in the solar array (wedge a) |
| Left transition panel | N/A | The panel to the left of a given ground position that defines the upper bound of wedge d in Fig 6a. Mathematically, it is the panel for which $\alpha_t < \phi$, and $\alpha_b > \omega$. |
| Right transition panel | N/A | The panel to the right of a given ground position that defines the upper bound of wedge d in Fig 6a. Mathematically, it is the panel for which $\alpha_t > \phi$, and $\alpha_b < \omega$. |

Their derivation is demonstrated in Fig 7b.

$$\alpha_b = \arctan\left(\frac{H_0}{D(i-1)-S}\right) \tag{9}$$

$$\alpha_t = \arctan\left(\frac{H_0 + L\sin\theta}{D(i-1) + L\cos\theta - S}\right) \tag{10}$$

If $\alpha_t < \phi$, and $\alpha_b > \omega$, then that panel is the *right* transition panel for that ground position. If the top or bottom angle is obtuse, that implies that the panel edge is to the left of the ground position, then the signs are reversed relative to $\omega$. In this case, if $\alpha_t > \phi$, and $\alpha_b < \omega$, then that panel is the *left* transition panel for that ground position. $\alpha_{t_L}$ and $\alpha_{t_R}$ are computed via Eq 14 when $i$ equals the index of the left and right transition panels, respectively. With the transition panels found, the wedges labeled with $d$ are known.

The final variable to determine is the angle of wedge $c$, which is the angle blocked by the panel(s) *between* the left and right transition panels (termed the "panel angle" in Fig 5).

The last consideration is the edge effects for ground positions underneath the leftmost or rightmost panel in the array. The relative magnitudes of the critical angles and the edge angles are used to identify edge cases. At the left edge of the array, the left edge angle is less than the left critical angle. Here, one shade wedge is eliminated (Fig 6b). The same applies for the right edge case, when the right edge angle is greater than the right critical angle.

The viewfactor, $F_v$, is computed for positions $S \in (0, D(N-1))$ in Eq 11.

$$F_v\{S\} = \frac{1}{\pi}\left(\pi - \alpha_{tL} - \alpha_{tR} + \mu + \gamma - \sum_i (\alpha_t - \alpha_b)_i\right) \tag{11}$$

(a)

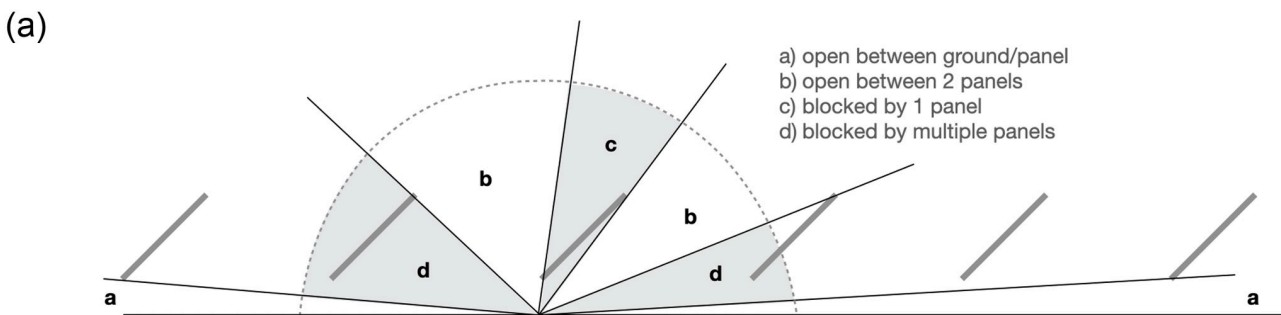

(b)

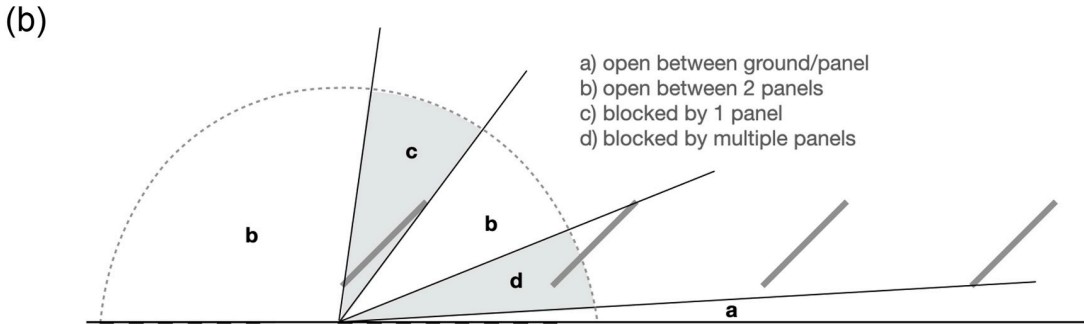

(c)

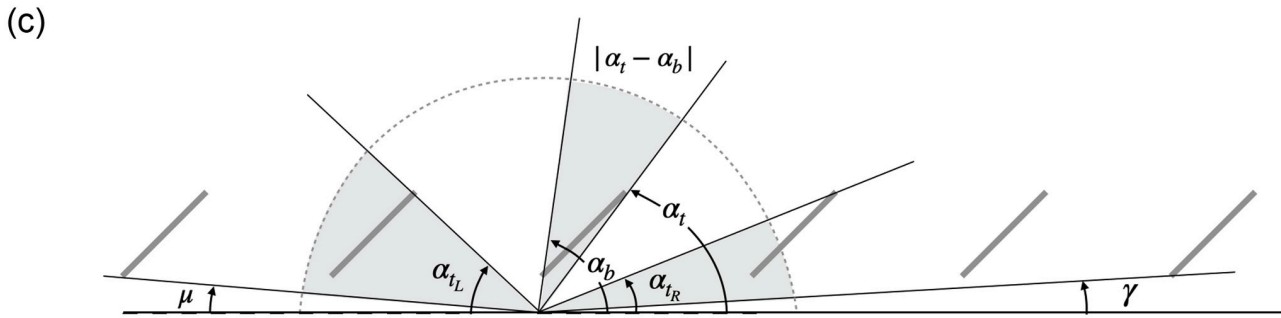

**Fig 6.** a. Wedges used to determine viewfactor. b. Wedges used to determine viewfactor in left edge cases. c. Angles used to determine viewfactor, variables labeled.

The viewfactor is a value between 0 and 1. $L_{sky}$ is scaled by the viewfactor to yield the amount of longwave radiation from the sky that $S$ receives.

Fig 6 is proportional to the dimensions of the case study array so that the angles could be verified by observation. Fig 6c illustrates how the wedges are defined by the variables referenced in the text.

The total flux, $q_{Tot}$ at each ground position is found by Eq 12.

$$q_{Tot}\{S\} = q\{S\} + F_V\{S\}L_{sky} \tag{12}$$

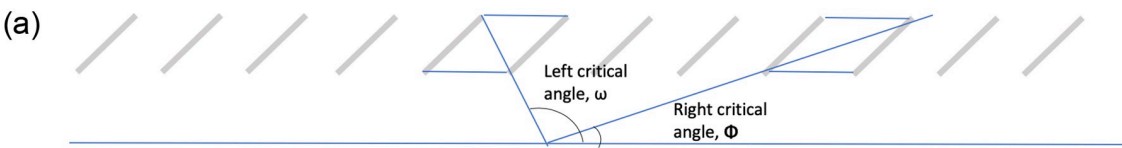

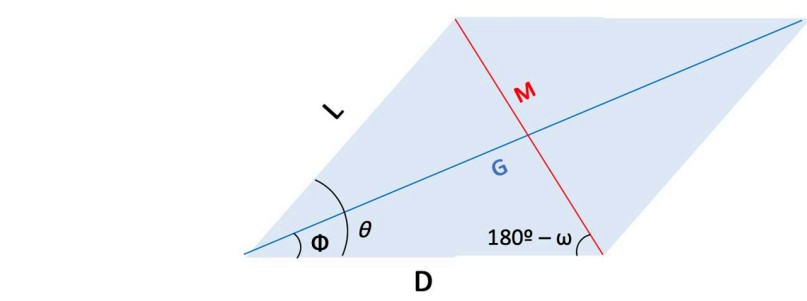

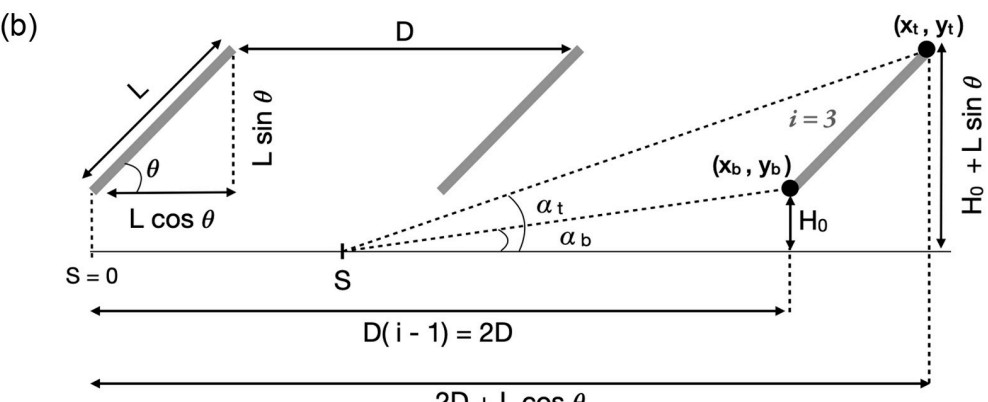

**Fig 7.** a. Derivation of critical angles for finding viewfactor. b. Derivation of top angle ($\alpha_t$) and bottom angle ($\alpha_b$) for panel $i = 3$.

### Flux over time: Determining panel temperature

The heat flux at the ground is not only dependent on position $S$, but also time $t$, as stated in Eq 13. $T_P$ and $L_{sky}$ are functions of air temperature, relative humidity, shortwave radiation, and wind speed. These parameters (displayed in Fig 8) were measured by Adeh et al. in 2019 [12] at the same solar array used in this case study, and integrated into our determination of panel temperature and longwave radiation from the sky.

$$q_{Tot}\{S,\ t\} = q\{S,\ T_P(t)\} + F_V\{S\} \cdot L_{sky}(t) \tag{13}$$

The panel temperature at a given time was found by performing an energy balance on the panel (Fig 9) and solving Eq 14 numerically for $T_P$. The longwave radiation from adjacent panels was not considered in the energy balance.

$$(1 - \alpha - \varepsilon)\,R_{sun} + L_{sky} + L_g - 2\sigma T_P^4 - 2h(T_P - T_{air}) = 0 \tag{14}$$

where $\alpha = 0.2$ is the panel albedo, $\varepsilon$ is the electrical efficiency of the panel, $R_{sun}$ is the measured

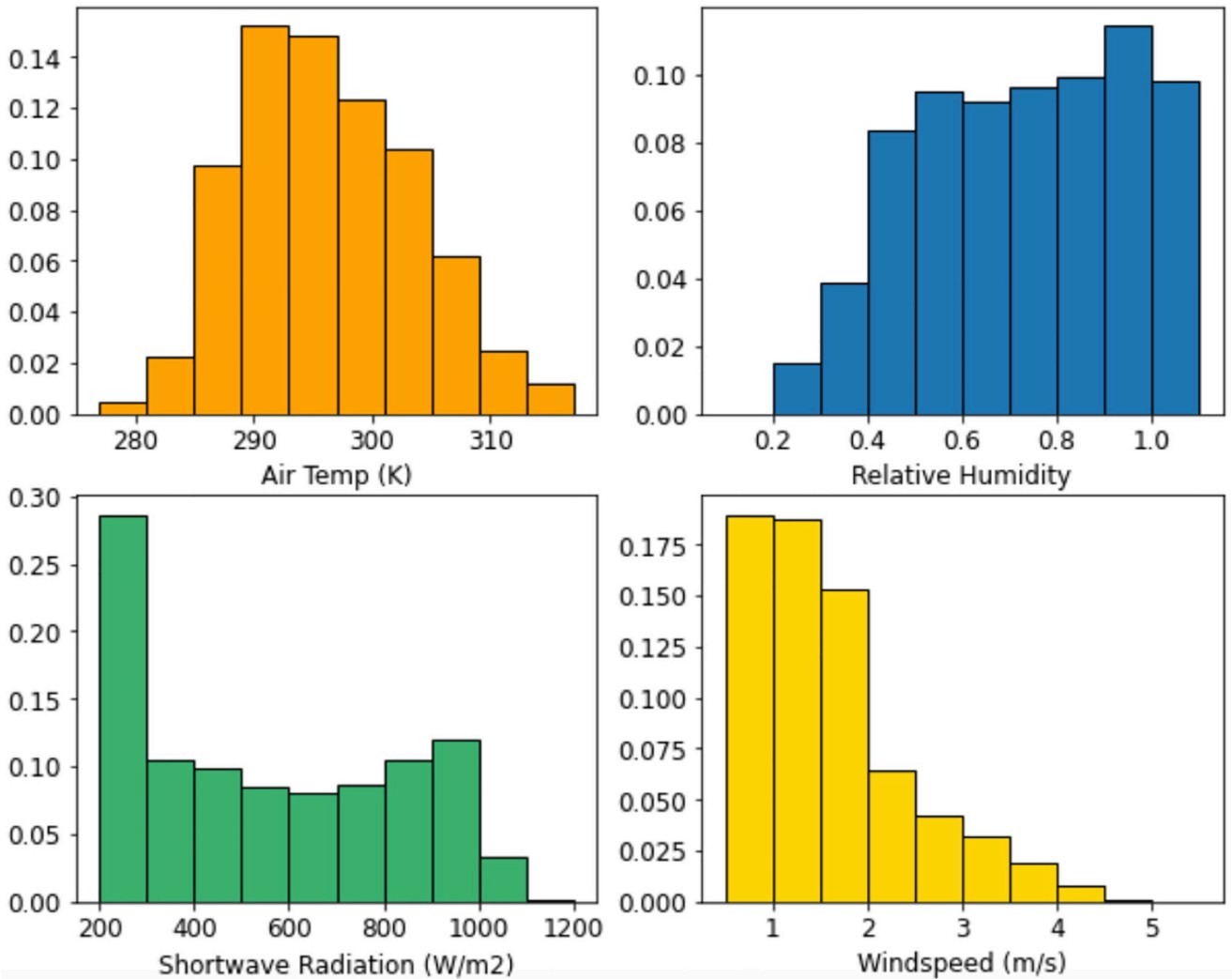

**Fig 8. Weather parameter data.**

incoming shortwave radiation at time $t$, $L_g$ is the longwave radiation from the ground, and $h$ is the convection coefficient of the air. The electrical efficiency is also a function of panel temperature:

$$\varepsilon = \varepsilon_{ref}\left[1 - 0.0051\left(T_P - T_{ref}\right)\right] \tag{15}$$

where $\varepsilon_{ref} = 0.135$ is the reference efficiency at $T_{ref} = 298$ K. This relationship is valid when $|T_P - T_{ref}| \leq 20$ K. $L_g$ is modeled by the black body radiation equation:

$$L_g = \sigma T_g^{\,4} \tag{16}$$

where $T_g$ is the temperature of the ground, and was assumed equivalent to $T_{air}$ at time $t$. The convection coefficient (W m$^{-2}$ K$^{-1}$) is described by:

$$h = 0.036\frac{k_{air}}{L}\left(\frac{uL_P}{v}\right)^{\frac{4}{5}}\text{Pr}^{\frac{1}{3}} \tag{17}$$

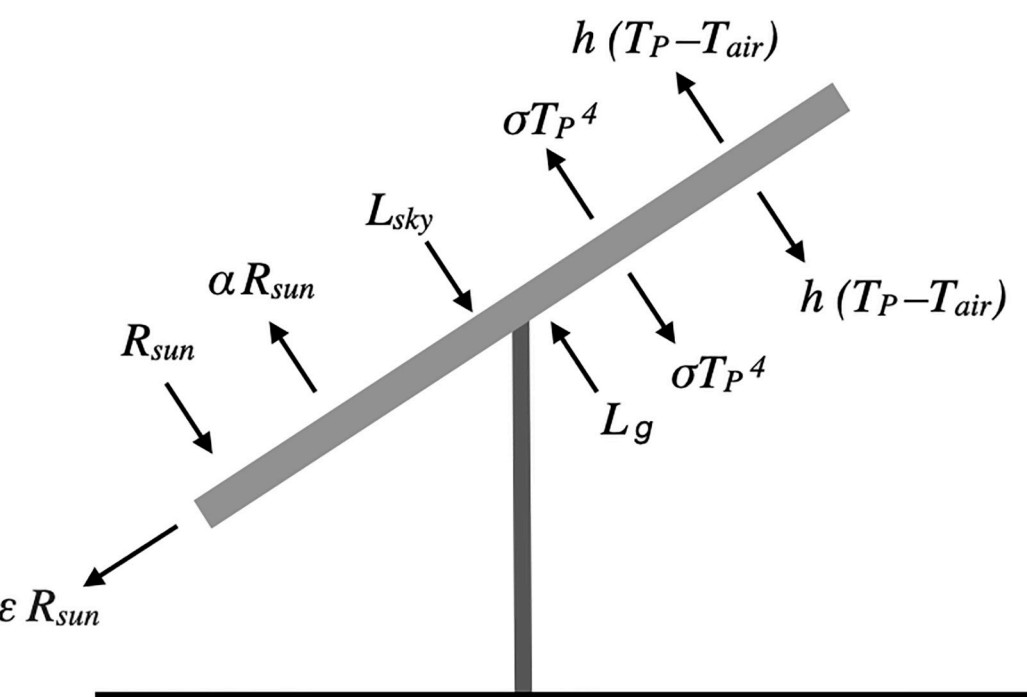

**Fig 9. Energy pathways for a panel, corresponding to Eq 14 (Adeh 2019).**

where $k_{air} = 0.026$ W m$^{-1}$ K$^{-1}$ is the thermal conductivity of dry air, $L$ is the length of the solar panel (m), $u$ is the measured wind speed (m/s) at the panel height at time $t$, $v = 1.57 \cdot 10^{-5}$ m$^2$/s is the kinematic viscosity of air, and Pr = 0.707 is the Prandtl number for dry air.

Finally, the longwave radiation from the sky, $L_{sky}$, was found for all $t$ using measured values. Fig 10 depicts the panel temperature and air temperature variation for a single day. Table 3 summarizes the inputs affecting panel temperature.

## Flux over time: Assimilating with spatial data

There were several gaps in the measured weather data due to instrument limitations. Small gaps (less than ~1 day) were filled in by interpolation. For larger gaps, the time step indices and corresponding days of the year were identified by inspection. Two large gaps existed, resulting in heat flux values that span the following three durations for the year 2018:

1. 00:00, May 6–11:00, May 22

2. 21:00, June 13–10:00, July 9

3. 17:00, July 13–00:00, August 28

The final step was to use the time-dependent panel temperature to determine the flux at each location and at each time step. This yielded the three contour plots in Fig 12, one plot for each timespan above. Finally, the flux was integrated across time to yield the total energy from longwave radiation accumulated over the entire season (not including the large gaps).

## Shortwave radiative heat flux

The shortwave radiative heat flux under the solar panels was computed and added to the longwave radiation to yield the total radiative heat flux at the land surface. The shortwave radiation

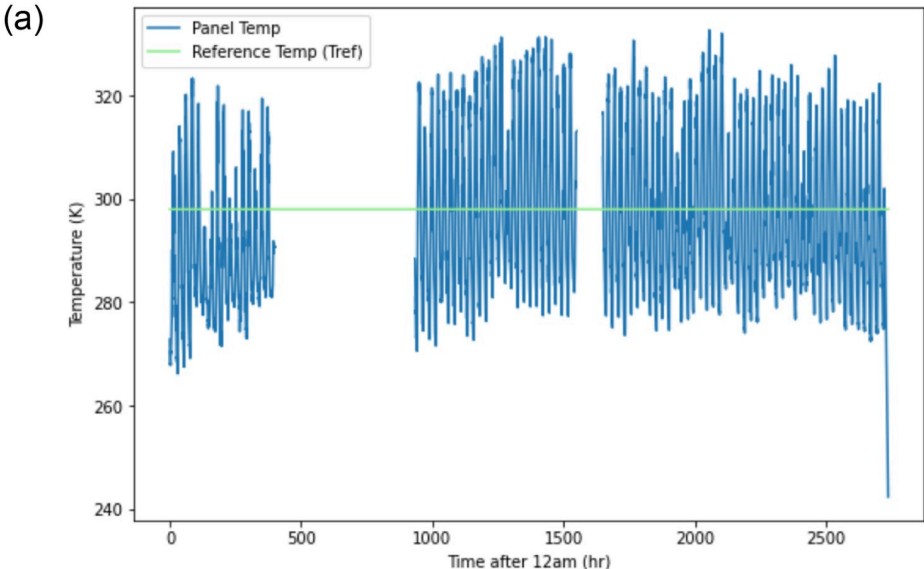

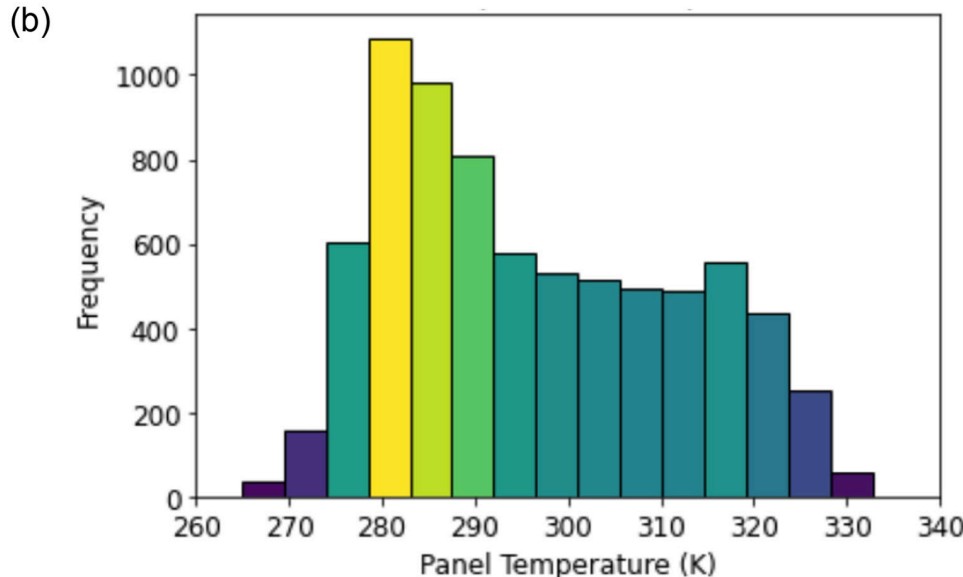

**Fig 10.** a. Panel temperature over entire season (6 May– 28 August). b. Histogram of panel temperature over entire season (6 May– 28 August).

was estimated for the three time spans using Autodesk's Revit 3D building information modeling (BIM) software. Revit's Insight Solar Analysis module uses the Perez solar model [13], which accounts for the geographical location and time of year when computing incoming solar radiation. It incorporates the direct normal irradiance (DNI) and diffuse horizontal irradiance (DHI). Revit's Dynamo visual programming language allows for the creation of parametrized 3D objects such as solar panels. The solar analysis accounts for solar array geometry, including panel height, panel length, tilt, array width, and distance between rows. Since the longwave radiation did not account for edge effects on the row ends, shortwave values were extracted from along the center of the array.

**Table 3. Parameters for panel temperature determination.**

| Variable Name | Symbol | Value | Units |
|---|---|---|---|
| Panel Albedo | $\alpha$ | 0.2 | - |
| Electrical efficiency of panel | $\varepsilon$ | Eq 15 | - |
| Reference efficiency | $\varepsilon_{ref}$ | 0.135 | - |
| Reference temperature | $T_{ref}$ | 298 | K |
| Incoming shortwave radiation | $R_{sun}$ | Measured | W m$^{-2}$ |
| Outgoing longwave radiation from ground | $L_g$ | Eq 16 | W m$^{-2}$ |
| Air temperature | $T_{air}$ | Measured | K |
| Convection coefficient of air | $h$ | Eq 17 | W m$^{-2}$ K$^{-1}$ |
| Thermal conductivity of dry air | $k_{air}$ | 0.026 | W m$^{-2}$ K$^{-1}$ |
| Kinematic viscosity of air | $v$ | $1.57 \cdot 10^{-5}$ | m$^2$/s |
| Prandtl number of dry air | Pr | 0.707 | - |
| Wind speed | $u$ | Measured | m/s |
| Downwelling longwave radiation | $L_{sky}$ | Eq 6 | W m$^{-2}$ |

## Results

Histograms of the measured air temperature, relative humidity, shortwave radiation, and wind speed are displayed in Fig 6 for reference.

Fig 10a displays the calculated panel temperature from May through August. Note the two long straight lines, which are interpolated values due to the absence of weather data collected during those periods. The reference temperature, $T_{ref}$ = 298 K, is a parameter in Eq 15 used to find panel efficiency. Recall that Eq 15 is most accurate when $|T_P - T_{ref}| \leq 20$ K. For the data in this case study, this difference is greater than 20 K for 15.8% of all panel temperatures. This limitation adds some degree of uncertainty to the panel temperature values.

Fig 10b illustrates the same panel temperatures as Fig 10a, but in a probability density function. The median value lies close to $T_{ref}$, and only the last three bars on each side are outside the 20 K difference requirement. The mode lies within 280–290 K (7 – 17ºC). Note that the panel temperature has a wider range than the air temperature in Fig 8.

Fig 11a illustrates the spatial variation and relative magnitudes of longwave radiation from the panels and sky at midnight on May 6. The vertical grey bars represent ground under the panels. The lower sinusoidal line is the longwave from the sky scaled by the viewfactor at each position. It is evident that at most positions, the longwave energy from the panels exceeds the maximum $L_{sky}$ value (i.e. what the control area in full sun receives). It is also clear that, from the sky only, all positions receive well below the maximum $L_{sky}$ value due to the viewfactor effect. The peaks in $L_{sky}$ received occur in the aisles of the array, while the troughs occur directly under the panels (with the exception of the ground positions beneath the two right-most panels).

Fig 11b takes a closer look at the longwave radiation from the panels, which is identical to the top line in Fig 11a. Moving in the positive $x$ direction from the peak of 360 W/m$^2$, one can see a gradual decrease in energy as the panel slopes up and away from the ground. However, the energy then spikes sharply upwards from about 290 W/m$^2$ to 380 W/m$^2$. Recall that the influence of each panel is bound by $x_{lim}$, which is a projection from the lower edge of a panel to the ground. The spike occurs to the immediate right of $x_{lim}$. For ground positons under panel 2 in Fig 11b, the spike corresponds to the transition between two positions: the position left of $x_{lim}$, which receives longwave energy from panel 2, and the position right of $x_{lim}$, which receives longwave energy from panels 2 *and* 3. Continuing to the right, the flux reaches a local minimum of 330 W/m$^2$ in the center of the aisle.

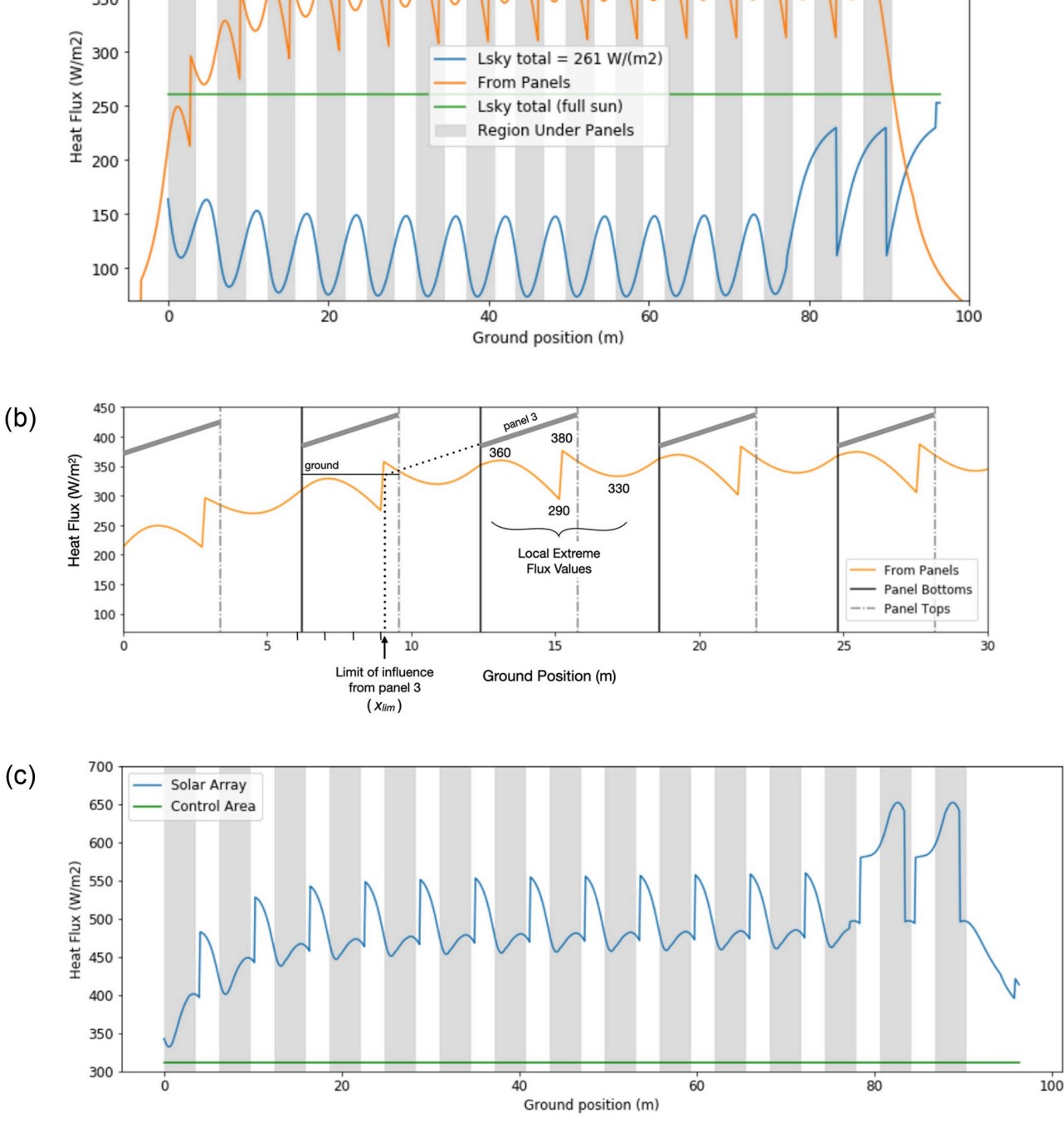

**Fig 11.** a. Components of the incoming longwave radiation at ground surface at t = 0 (May 6, 12am). b. Longwave from panels at t = 0 (May 6, 12am). c. Total incoming longwave radiation at ground surface at t = 0 (sum of the two lines in Fig 11a).

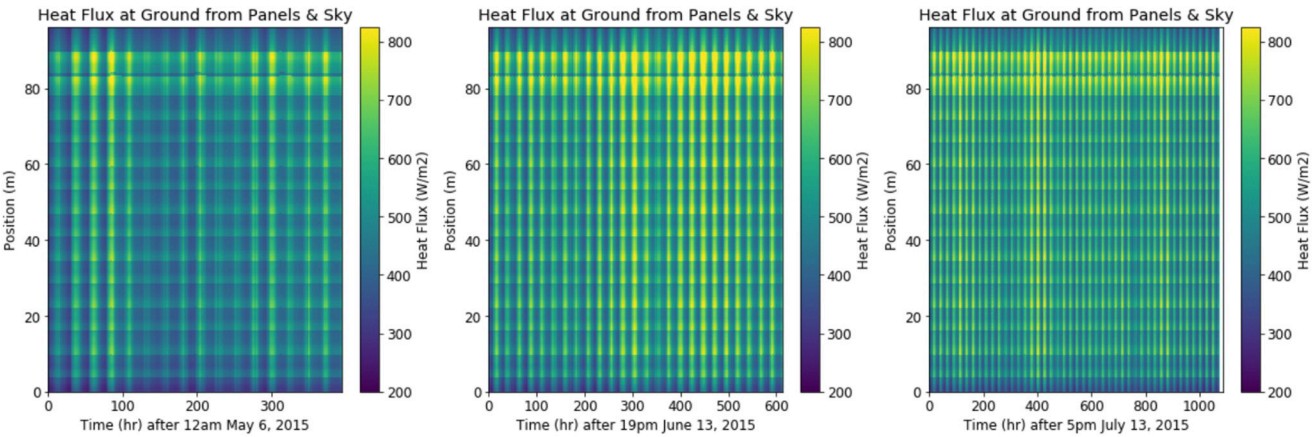

**Fig 12. Plots of heat flux over space and time.**

The total incoming longwave radiation at the ground surface is displayed in Fig 11c. The resulting maxima occur to the immediate right of the upper panel edges. It may seem counter-intuitive that the peaks in total longwave energy do not occur under the panels, where the ground is closest to the panel. However, under the panels, there is less exposure to sky (i.e. the viewfactor is smaller, so there is less longwave energy received from the sky). This effect offsets the higher longwave energy received from the panels. At all locations in the array, the total heat flux exceeds that of the control area.

Fig 12 captures the spatial and temporal variation in longwave heat flux at the ground for all three timespans. The vertical axis represents ground position. Variability in this axis arises from the 15 panels identical to our case study. The horizontal axis represents time. Variability along this axis comes from diurnal variability in weather data. For example, at the hottest point in a day, the heat flux ranges from about 600–800 W/m$^2$. At night, the heat flux drops as low as 200 W/m$^2$. Seasonal variation is also apparent, with higher heat fluxes in June and July (center plot). Highest heat flux values occur at the north edge of the array (~40–80 m). This is the same location as the highest viewfactors, and therefore the highest longwave heat flux from the sky, as seen in Fig 8a.

Fig 13 is the product of four steps: (1) Fig 12 is integrated across time to yield the total long-wave energy accumulated over the season (13a), (2) the Revit model predicts the shortwave radiation energy accumulated over the season (Fig 13a), and (3) we sum these two data sets to yield the total incoming energy at the ground under the panels across the season (Fig 13b). The flat line in all plots represents the radiative heat flux at the control plot in full sun. In Fig 13a, the difference between the shortwave radiation in the open aisles (white areas) and the control arises (blue dotted line) is from low sun angles. The large shadows that occur during low sun angles contribute to this difference.

Finally, Fig 14 portrays the percentage of the total incoming heat flux that is contributed by longwave radiation. In the open aisles, where shortwave is high, longwave makes up ~5% of the total incoming heat flux. Directly under the panels, where shortwave is low, longwave makes up ~70% of the total.

## Discussion & conclusions

Notably, the longwave energy at all ground locations under the solar panels is much higher than the longwave energy in the control area. When shortwave is considered, the longwave

(a)

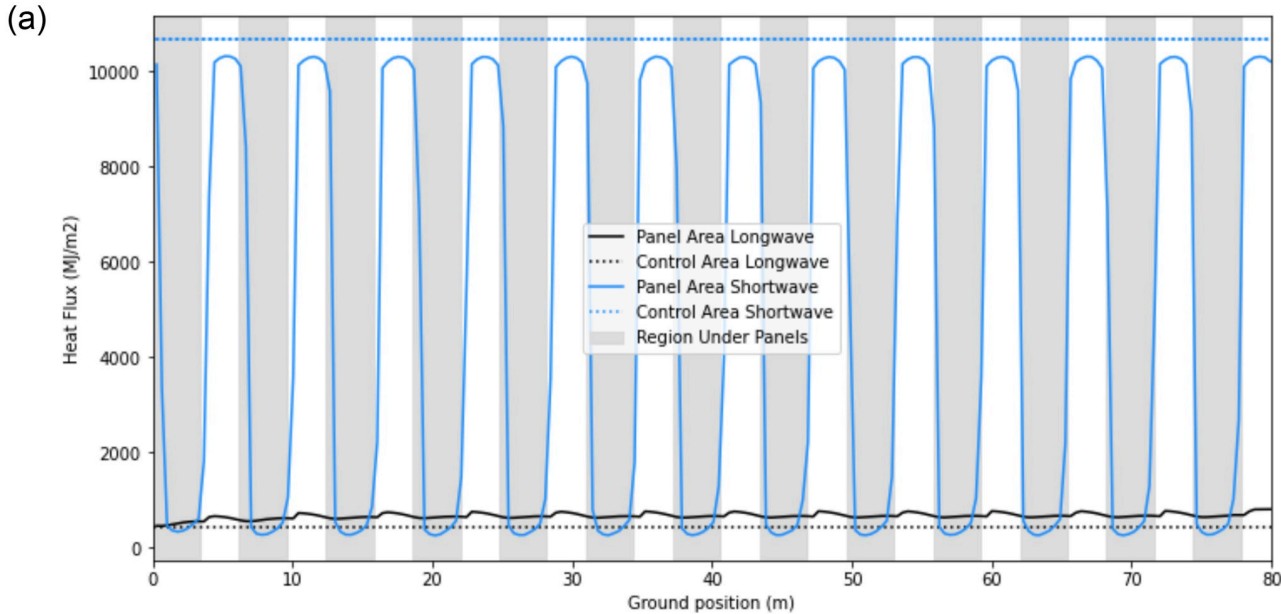

(b)

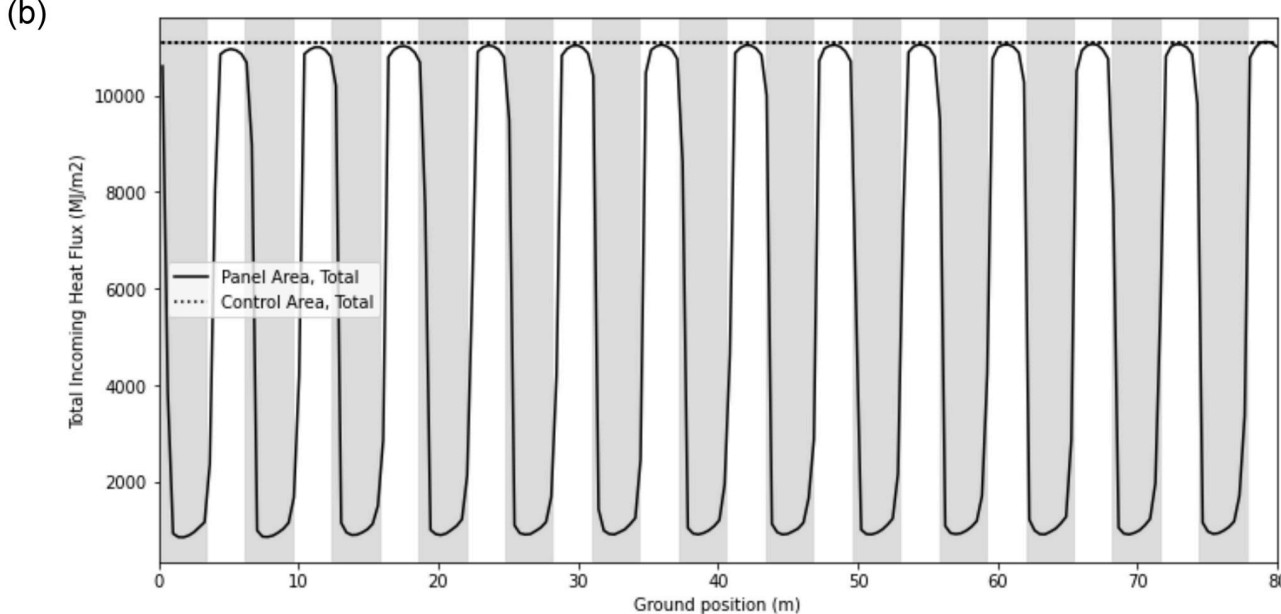

**Fig 13.** a. Longwave and shortwave radiation at ground under panels vs. full sun (control), accumulated over May 6 to May 22, 2015. b. Total incoming radiation at ground under panels vs. full sun (control), accumulated over May 6 to May 22, 2015.

heat flux is ~5% of the total incoming radiation at the peaks, and 70% of the total at the troughs (Fig 14). This suggests that even in the open aisles where sunlight is a major source of energy, longwave radiation can meaningfully impact the local energy budget. Indeed, the model demonstrates that longwave energy should not be neglected when considering a full energy balance on the soil under solar panels.

The net downwelling radiation in the open aisles is the same as the control, but there's less usable energy for plants. This raises an interesting question: if the available energy at the land

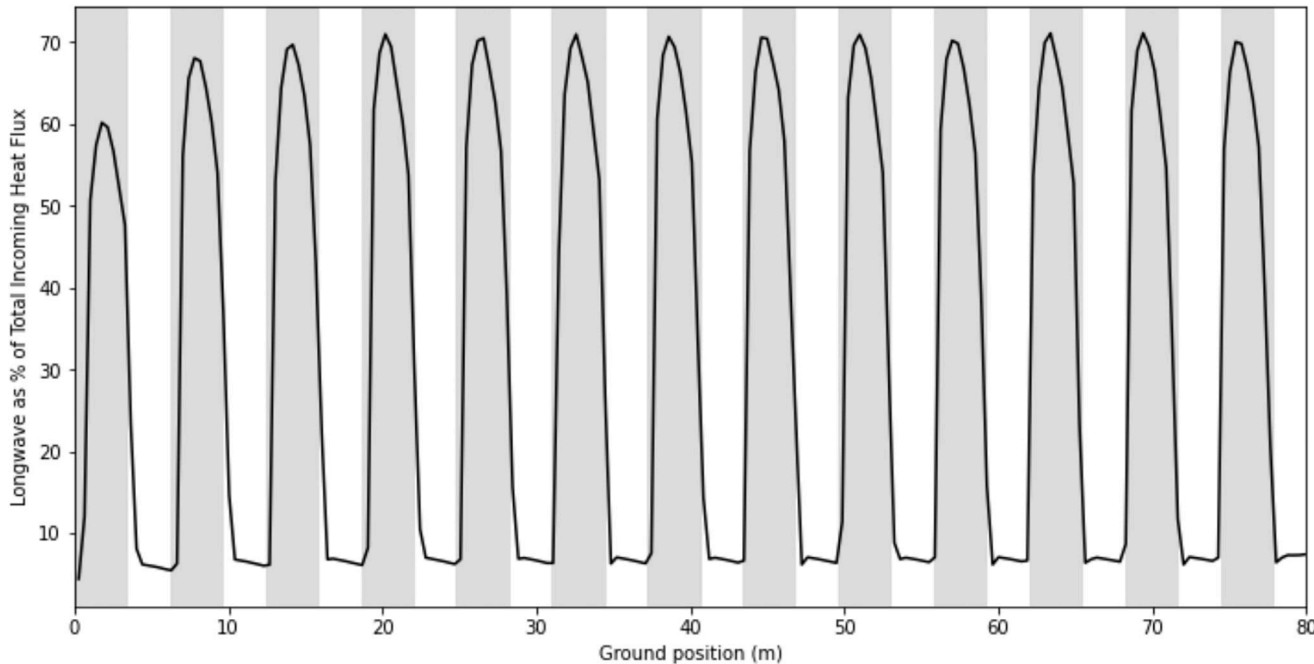

**Fig 14. Longwave radiative heat flux as a percentage of the total incoming heat flux (Fig 13b) for all ground positions.**

surface is the same, then will the other energy fluxes follow suit? Field observations suggest otherwise [11]. Thus, we infer that energy partitioning is impacted.

The model predicts locations and intensities of microclimates based on array geometry and local climate. It could be manipulated to optimize parameters for a solar array. For example, the model could be reversed to output panel geometry based on the energy and water needs for a certain crop. Alternatively, it could be used to determine the panel geometry that minimizes the spatial differences in heat flux through the ground surface. This may enhance uniformity in soil moisture and crop performance. Future studies should adapt this work into a full energy budget analysis; this model may be combined with models that predict all other sources of energy into or out of the soil, such as evaporative cooling, outgoing longwave radiation, etc. This would enable estimates of the total energy available for sensible, latent, and ground heat flux.

The model results were verified to be within physical and systemic bounds. Total longwave energy under the panels is always greater than zero and less than the *ad extremum*: the heat flux value that would occur if the entire area was covered by solar panels. Viewfactors were between zero and one, with zero as the *ad extremum* case and one as the control case. They were also verified against direct measurements on to-scale diagrams.

The model's assumptions produce some limitations. First, it neglects edge effects along the western and eastern boundaries, so the ground surface at theses edges of the array will experience different heat flux values than those predicted. The model also predicts panel temperature based on an energy balance, but it would be more accurate to measure it directly. The most significant assumption in the energy balance is that longwave energy from other panels is negligible; if the longwave energy of a panel produces significant effects on the heat flux through the ground, then it should also affect the temperature of nearby panels. Accounting for this would increase the resulting heat flux at the ground. We also assumed that the panels were black

bodies (i.e. have an emissivity of 1), but the actual emissivity, *E*, is less. Using the actual emissivity would decrease the resulting heat flux values by the factor *E*.

Another approximation in the model is the downwelling longwave radiation from the sky, which assumes clear skies and takes the empirical inputs of relative humidity and air temperature. A more precise estimation of the downwelling radiation would use additional inputs of surface air pressure, the e-folding height-scale of water vapor, and the $CO_2$ concentration [14]. However, the clear skies assumption is the case where longwave radiation has the minimum contribution to the energy balance. That is, high shortwave radiation and low downwelling longwave radiation. Note that in Fig 14, the longwave varies from ~5% to 70% of the total incoming radiation. Therefore, even in this case, it cannot be ignored. It is critical to consider longwave radiation when analyzing agrivoltaic systems, especially in arid regions where clear skies are prevalent, and where agrivoltaics have their highest potential.

## Supporting information

**S1 File.**
(PDF)

## Acknowledgments

This paper was inspired by the work of Elnaz Hassanpour Adeh, whose agrivoltaics research revealed the anomaly in soil moisture between ground under soil panels and ground in full sun. Her weather data was inputted into the model, providing a real case study to simulate.

## Author Contributions

**Conceptualization:** Chad W. Higgins.

**Formal analysis:** Laurel A. Shepard.

**Investigation:** Laurel A. Shepard.

**Methodology:** Laurel A. Shepard, Chad W. Higgins, Kyle W. Proctor.

**Software:** Laurel A. Shepard, Kyle W. Proctor.

**Supervision:** Chad W. Higgins.

**Visualization:** Laurel A. Shepard.

**Writing – original draft:** Laurel A. Shepard.

**Writing – review & editing:** Chad W. Higgins, Kyle W. Proctor.

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
