## [Decision Letter · Decision Letter 0]

31 Mar 2022

PONE-D-22-06438Agrivoltaics: Modeling the effect of longwave radiation from solar panels on soil moisturePLOS ONE

Dear Dr. Shepard,

Thank you for submitting your manuscript to PLOS ONE. After careful consideration, we feel that it has merit but does not fully meet PLOS ONE’s publication criteria as it currently stands. Therefore, we invite you to submit a revised version of the manuscript that addresses the points raised during the review process.

We look forward to receiving your revised manuscript.

Kind regards,

Domenico Mazzeo

Academic Editor

PLOS ONE

Journal Requirements:

Reviewers' comments:

Reviewer's Responses to Questions

**Comments to the Author**

1. Is the manuscript technically sound, and do the data support the conclusions?

Reviewer #1: Yes

Reviewer #2: Yes

2. Has the statistical analysis been performed appropriately and rigorously? 

Reviewer #1: Yes

Reviewer #2: Yes

3. Have the authors made all data underlying the findings in their manuscript fully available?

Reviewer #1: Yes

Reviewer #2: Yes

4. Is the manuscript presented in an intelligible fashion and written in standard English?

Reviewer #1: Yes

Reviewer #2: Yes

5. Review Comments to the Author

Reviewer #1: The work is logically structured and consequential in its exposition. The article develops a model to quantify the downwelling longwave energy at the ground to explore the soil moisture loss under the time of 2018. I cannot recommend considering this paper for publication before major modifications and change. My comments are as follows:

1. The order result of uploaded image is a mess. In addition, the horizontal and vertical coordinates are not clear in Fig. 3.

2. The missing weather parameters for small time steps (less than 1 day) were approximated via interpolation in this paper, but the larger time steps (more than 1 day)?

3. The clear skies condition is assumed in modeling work, whether the proposed model can be set up under more complex environmental conditions (cloudy sky, cloudy in the morning and clear sky in the afternoon). All measured data was collected in the clear skies in the model? Please provide detail explain the applicability and advantages of model.

4. Two long straight lines describe the interpolated values as shown in Fig. 10a, the first straight line shows a long time. However, it shows a volatility change, how to verify the rationality of a long time interpolation, data consistency or data series test?

5. The frequencies of panel temperatures show an obvious frequency difference in Fig. 10b, please explain this change.

6. In the work I do not see any mention about model both parameter calibration and validation. A model is applied in this study, it is suggested that these differences of simulated and measured result can be emphasized the quantified analysis.

7. I believe that the Authors should do more work. The study analyzes the difference and change of in longwave energy, and another approximation of downwelling longwave radiation in the model is computed. It should provide the more discusses and analysis for the effect of longwave radiation on soil moisture.

Reviewer #2: This is a interesting research dealing with the calculation of the long-wave radiation flux induced by the solar panels. The authors presented detailed and analytical geometrical schemes and calculating procedures, which provide very good insights for future research. However, the problem is that, are the data of calculated long-wave radiation flux right? why the evaporation contributed by it could be 60-80 cm, and you admitted that the value was too high. There might be something wrong in calculation the radiation, or the calculation of evaporation. I expect more solid and useful results, that should be, your calculation of the ground intercepted radiation and ground soil evaporation are all validated by the field measurements. Both long and short waves should be considered.

6. PLOS authors have the option to publish the peer review history of their article (what does this mean?). If published, this will include your full peer review and any attached files.

Reviewer #1: No

Reviewer #2: No

---

## [Author Response · Author response to Decision Letter 0]

28 Jun 2022

All comments can be found in our Response to Reviewers document.

---

## [Decision Letter · Decision Letter 1]

3 Aug 2022

Agrivoltaics: Modeling the relative importance of longwave radiation from solar panels

PONE-D-22-06438R1

Dear Dr. Shepard,

We’re pleased to inform you that your manuscript has been judged scientifically suitable for publication and will be formally accepted for publication once it meets all outstanding technical requirements.

Kind regards,

Domenico Mazzeo

Academic Editor

PLOS ONE

Additional Editor Comments (optional):

Reviewers' comments:

Reviewer's Responses to Questions

**Comments to the Author**

1. If the authors have adequately addressed your comments raised in a previous round of review and you feel that this manuscript is now acceptable for publication, you may indicate that here to bypass the “Comments to the Author” section, enter your conflict of interest statement in the “Confidential to Editor” section, and submit your "Accept" recommendation.

Reviewer #1: All comments have been addressed

2. Is the manuscript technically sound, and do the data support the conclusions?

Reviewer #1: Yes

3. Has the statistical analysis been performed appropriately and rigorously? 

Reviewer #1: Yes

4. Have the authors made all data underlying the findings in their manuscript fully available?

Reviewer #1: Yes

5. Is the manuscript presented in an intelligible fashion and written in standard English?

Reviewer #1: Yes

6. Review Comments to the Author

Reviewer #1: The questions listed before are all addressed and I have no further comments for this paper. My suggestion to this paper is accepted for publication.

7. PLOS authors have the option to publish the peer review history of their article (what does this mean?). If published, this will include your full peer review and any attached files.

Reviewer #1: No

---

## [Editor Report · Acceptance letter]

10 Oct 2022

PONE-D-22-06438R1 

Agrivoltaics: Modeling the relative importance of longwave radiation from solar panels 

Dear Dr. Shepard:

I'm pleased to inform you that your manuscript has been deemed suitable for publication in PLOS ONE. Congratulations! Your manuscript is now with our production department. 

Kind regards, 

on behalf of

PhD Domenico Mazzeo 

Academic Editor

PLOS ONE